

# Fluid-rock interaction in the intraplate active seismic zone: Boon or bane?

**Piyal Halder[1,2], Matsyendra Kumar Shukla[3], Kamlesh Kumar[1,2], Anupam Sharma[1,2*]**

[1]Birbal Sahni Institute of Palaeosciences (DST, Govt. of India), Lucknow-226007, Uttar Pradesh, India
[2]Academy of Scientific and Innovative Research (AcSIR), Ghaziabad- 201002, India
[3]Govt. of India, Ministry of Earth Sciences (MoES), Borehole Geophysics Research Laboratory (BGRL), Karad-415105, India

*Correspondence to:* Anupam Sharma (anupam110367@gmail.com; anupam.sharma@bsip.res.in)

**Abstract**

The Koyna-Warna Seismogenic Region of western India has been recognized as one of the hotspots for reservoir-triggered seismicity (RTS) since 1967. The current study investigates the fluid's interaction with the severely fractured granitoid basement of this area and its potential contribution to the recurring seismicity. The presence of several secondary minerals, such as chlorite, epidote, calcite, illite, etc., along the pre-existing faults and fractures, is revealed by detailed petrologic investigation at mesoscopic and microscopic scales along with XRD

analysis. This indicates the fluid-rock interaction along these mechanically weak planes and subsequent propylitic grade of hydrothermal alteration under acidic to neutral conditions (pH 5.5-7) and the temperature of above 200-220 °C up to about 350 °C. Additionally, the transformation of biotite into chlorite due to fluid interaction has been inferred from the microscopic appearance of biotitic remnant within neoformed chlorite which is further supported by the mass loss of $K_2O$ and concurrent gain of MgO and FeO, demonstrating the replacement of

potassium (K) interlayer sheet by brucite-like [Mg (OH)$_2$] layer during biotite chloritization. However, this released $K_2O$ further assists in the formation of illite resulting in the mass gain of $K_2O$ at a few certain depths, whereas the dissolution of plagioclase justifies the formation of albite and calcite as evidenced by the gain of $Na_2O$ and CaO. The present study also highlights that the recurring nature of the seismicity in this area may be related to clay mineralization along the faults and fractures due to fluid-rock interaction, such as chlorite, illite,

etc., in addition to the existing fault geometry and stress build-up due to reservoir impoundment. At increasing stress condition, the anisotropic and weakly bonded, layered crystal structure of chlorite forming ripplocations may develop kink bands and increases the yield strength proportionally with rising pressure up to dehydration temperature. Such visco-elastic behaviour of chlorite may promote aseismic creep in the faults. On the other hand, epidote noticed at certain depths has a contrasting behaviour; it tends to wear at the micron or submicron-scale

asperity contacts and produce fine particles which generate unstable sliding. However, the relatively higher abundance of chlorite in the faults and fractures disrupts the epidote-epidote contact asperities and prevents such wearing of epidote grains into fine particles. Thus, biotite chloritization in conjunction with relatively less production of epidote along pre-existing faults and fractures helps to release the accumulated stress through a series of small-scale earthquakes and results in the steady fault creep observed in this region during the past 50

years. In this context, fluid-rock interaction along the pre-existing faults and fractures at shallow depth has acted as a blessing for the Koyna-Warna Seismogenic region shielding it from relatively large magnitude earthquakes – a boon for the region.





## 1 Introduction

Faults and fractures act as fluid migration pathways during seismic rupture as well as the inter-seismic creep periods (Duan et al., 2016). In order to achieve the thermodynamic equilibrium, this infiltrated fluid interacts with the host rocks and chemically reacts with the constituent minerals. As a result, a set of new minerals are formed that are stable under the newly attained conditions. However, the entire process of fracture-scale fluid-rock interaction basically occurs through the dissolution-transportation-precipitation (DTP) mechanism (Glassley et

al., 2016). When the percolating fluid is undersaturated in a certain mineral phase, dissolution primarily happens with unstable minerals, whereas precipitation occurs when the fluid is oversaturated with a particular mineral phase. On the other hand, transportation governing the flow of mass and heat occurs mainly in two ways- Diffusion and Advection (Bons et al., 2012; Glassley et al., 2016).

Diffusion encompasses the small-scale (<1 m) movement of the substance or elements from high concentration

to low concentration and thus involves mass gain (Bickle and Mckenzie, 1987). For example, the migration of calcium from a calcium-rich or calcareous host rock into other leads to the formation of calcite veins resulting in mass gain. Therefore, diffusion results in the mass flow following the concentration gradient in any chemical activity. Similarly, chemical disequilibrium created by the variation in pressure between the fluid in the fractures and the host rock may lead to the transportation of elements along fractures.  On the other hand, advection is

another method of mass flow through fractures, in which elements or substances from outside are carried by a fluid over the fracture networks, through the pore spaces, or along grain boundaries to the dilatation site of an open system. However, advection can transport elements over crustal-scale distances as being controlled by variations in the hydraulic head (Bons et al., 2012). Advection can also aid in hydrothermal alteration by transferring heat along pre-existing fractures in a horizontal fashion. Thus, advection results in the flow of mass

as well as heat during hydrothermal alteration. Besides, the DTP mechanism, the ion exchange mechanism is another way of fluid-rock interaction, which involves the exchange of ions between water and crystallographic sites in the mineral.

In addition to chemical and mineralogical changes due to the flow of mass and heat, the fluid-rock interaction has a significant impact on the mechanical and petrophysical properties of the host rocks particularly in the fault

zones. The frictional properties of the neoformed secondary minerals in the fault zones may control the slip behaviour of the faults and vice versa. Minerals with low frictional strength such as smectite facilitate the seismic slip while minerals with high frictional strength like chlorite promote creep. On the other hand, an increase in fault slip rate may result in a decrease in steady-state frictional strength, called velocity-weakening behaviour, whereas the reverse frictional response with a decrease in fault slip rate is termed velocity-strengthening behaviour

(Roesner et al., 2022). Velocity-strengthening behaviour on faults tends to inhibit earthquake nucleation and favours steady creep but velocity-weakening behaviour promotes earthquake nucleation and facilitates slip typical of earthquakes. Thus, fault or fracture-scale fluid-rock interaction may play a key role in fault mechanics as also evidenced in the 3 km deep fault rocks along the central creeping part of the San Andreas Fault (SAF) (Bradbury et al., 2015). Therefore, fluid-rock interaction and its effects on the seismic cycle are equally crucial for

comprehending the fault mechanics and the potential for future ruptures along faults in any seismically active region, even in an intraplate zone.



In the western part of the Indian subcontinent, five years after the impoundment of the Shivajisagar Reservoir behind the Koyna Dam in 1967, the Koyna-Warna Intraplate Region was devastated by an earthquake of magnitude 7.5 on the Richter Scale (according to the Indian Meteorological Department and the Central Water

and Power Research Station). Nearly 200 people lost their lives as a result of this catastrophic earthquake with the injury of more than a thousand up to 700 kilometres away from the epicentre, which was located only within 5 km of the Koyna Dam (Narain and Gupta, 1968). Since 1967, more than 100,000 earthquakes of magnitude greater than 1.0, about 200 earthquakes of magnitude around 4 and 22 earthquakes of magnitude greater than 5 have taken place in this region (Das and Mallik, 2020). The increase in the number of seismic events has been observed when

the water level in the reservoirs is at its minimum (Gupta, 2001), which indicates that the stress is accumulated during the infilling of the reservoir along the pre-existing faults and fractures and it gets released when the hydrostatic pressure from the top is reduced due to draining of water. Such commensurate correlation between the water level in the reservoirs and the frequency of earthquakes allowed the researchers to recognize this seismicity as the artificial-reservoir-triggered-seismicity (RTS) in which the stress drop associated with earthquakes is

accounted for by the causative activity (Gupta, 2002). In the Koyna-Warna Region, stress drop related to earthquakes is much larger than the stress level changes due to water infilling in the reservoirs (Gupta, 2002). However, the periodicity in seismic energy release has been found with a delay of a month after the annual filling and draining of the reservoirs, until 1996, which can be attributed to the diffusion of reservoir water through the pre-existing faults and fractures that raise the pore fluid pressure of the critically stressed medium and facilitates

earthquakes (Pandey and Chadha, 2003)

Therefore, to understand the impact of fluid flow through the pre-existing fractures or faults on the chemical and physical properties of the basement rocks of the Seismogenic region, this study has been initiated on the core samples, recovered by the Scientific Deep Drilling Program. Based on the mesoscopic, and microscopic evidence as well as supporting geochemical studies in the basement rocks of this area, the current article discusses the

mechanism of secondary mineralization in terms of the gain and loss of mass caused by fluid-rock interaction. The study also sheds light on how secondary minerals may bring changes in the physical properties of rocks resulting in the occurrence of earthquakes in the Koyna-Warna Seismogenic Region. Therefore, the outcomes of this paper are important not only in the context of Indian RTS (Reservoir Triggered Seismicity) but also in light of other instances of fluid-rock interaction that have been observed in various fault drilling projects around the

globe including SAFOD (Bradbury et al.), Nojima fault (Japan), Chelungpu fault (Taiwan), Aigion, Helike and Pyrgaki Faults in the Gulf of Corinth, among others.

## 2 Study Area

### 2.1 Lithology

The study area, Koyna-Warna Seismogenic Region, is located in the western part of the Indian shield over the

~65 Myr old Deccan traps province across the Western Ghat escarpments **(Fig. 1a)**. The Deccan basalt of Western Ghats has been divided into three Subgroups: Kalsubai, Lonavala and Wai (Beane et al., 1986). The amygdaloidal compound lava flow of the Kalsubai Subgroup is composed of phenocrysts of olivine and clinopyroxene and it has been subdivided into five different formations- Jawahar, Igatpuri, Neral, Thakurwadi and Bhimshankar. The Kalsubai subgroup is overlain by the Lonavala Subgroup consisting of the Khandala Formation and the Bushe



Formation. Below the Lonavala Subgroup, the Wai Subgroup composed of simple as well as compound flows with well-developed phenocrysts of plagioclase, pyroxene and olivine is subdivided into Poladpur, Ambenali, and Mahabaleswar Formations. The Deccan trap segment of the study area basically falls under this Wai Subgroup which is directly underlain by the granitoid basement rocks without any traces of infratrappean sediments in between (Gupta et al., 2015; Roy et al., 2013).



**Figure 1.** Geological and Seismologic Map of the Koyna-Warna Seismogenic Region. (a) The lithologic map of the Deccan Trap Volcanic Province of Maharashtra shows the areal distribution of the Koyna-Warna Seismogenic



Region (Rectangular box). The subsidiary zoomed-out physiographic map shows the position of the studied borehole (KBH1) with respect to the Koyna and Warna Reservoirs as well as the Koyna River Fault Zone (KRFZ),

Patan fault and NW-SE trending fractures (modified after Sinha et. al., 2017 and Rao et. al, 2017). (b) The seismicity map shows the earthquakes that occurred from 1967 to 2017 along with the focal mechanisms of the 1967 M 6.3 and 2012 April M 4.8 earthquakes and the adjacent diagram represents the longitude-wise distribution of the Focal depths of the earthquakes that occurred during 2011–2017. The dashed line indicates the trace of the Donichawadi fissure zone formed during the 1967 M6.3 Koyna earthquake (modified after Modak et.al., 2022).

130       The basement rocks of Neoarchean age (~2700 Ma) comprising predominantly of cratonic (tonalite-trondhjemite-granodiorite) gneisses of peninsular India and they are found as interlayers of granite, granite gneiss, and varying proportions of migmatitic gneiss (Goswami et al., 2017; Misra et al., 2017b; Shukla et al., 2022). The interlocking arrangement of the constituent mineral grains in granite portrays its igneous origin whereas in the granite gneiss section felsic and mafic minerals are visible in different dark and light-coloured alternate bands.

However, the leucosomes containing felsic minerals are primarily found as veins within mafic mineral-rich melanosomes in migmatitic gneiss sections, indicating that they are formed by the partial mixture of liquid produced during metamorphism and then separated as veins (Misra et al., 2017b).

**2.2 Structural features and stress pattern**

Since the impoundment of Shivajisagar Reservoir behind the Koyna Dam in 1962, this region has been regarded

as one of the most important sites of artificial-reservoir-triggered Seismicity (RTS). Later, the establishment of the Warna Reservoir in 1985 resulted in an enhancement in the frequency of earthquakes (Gupta, 2002). This Seismogenic zone spans an area of 600 $Km^2$ between the Koyna reservoir in the north and the Warna reservoir in the south, covering both sides of the Western Ghats escarpment (Goswami et al., 2017). The region is structurally bounded by the NE-SW trending Patan fault in the east and the Koyna River fault zone in the west (Fig. 1a)

(Talwani, 1997b; Talwani, 1997a). Besides, several NW–SE trending fracture planes and numerous small-scale faults are present in the region (Talwani, 1997b). The fault planes have been found orienting NNE with a dip of ~ 66° towards the west (Das and Mallik, 2020). The Donichawadi fissure zone, an NNE-SSW surface rupture zone with many en-echelon type fractures, near vertical fissures, and oblique or diagonal tensional cracks, has also been found close to the Kadoli hamlet (Modak et al., 2022; Misra et al., 2017a). This 200 km long and 4 km

broad fissure zone extending up to 10 km depth has been active since the 1967 earthquake and has been recognized as the surface manifestation of the permeable Donichawadi fault (Modak et al., 2022)

The majority of the earthquakes in the Koyna Seismogenic zone are governed by strike-slip faults with some normal components and in the Warna seismic zone by the normal faults (Goswami et al., 2020; Rao and Shashidhar, 2016). Focal mechanism solutions have revealed a unique alternate occurrence of strike–slip and

normal faulting instead of the dominance of any single component (Rao and Shashidhar, 2016). During 1967-1973, there was the predominance of the strike–slip faulting when the horizontal tectonic stress build-up reached the threshold along the NE-SW oriented faults. During 1973-1989 a transition into normal faulting was noticed NNW oriented near-vertical fault planes due to gravitation loading, which is possible only after the complete release of the horizontal stress accumulated along the NE-SW trending faults. This complete release of the

horizontal stress might have occurred due to the reservoir impoundments. However, the build-up of this horizontal



stress upto such a critical level can't be explained only by the RTS mechanism rather it was a combined effect of the reservoir impoundment and N–S compression of the Indian plate (Gahalaut et al., 2004). A few more changeovers in faulting mechanisms have also been seen with similar time spans, corresponding to the periods 1989–1995, 1995–2011 and onwards (Rao and Shashidhar, 2016). The normal faulting phases were of longer

duration (~16 years) in comparison to the phases of strike-slip faulting, which indicates that the release of stress due to the reservoir impoundments is faster than the stress built-up as the stress transmission from the plate boundary into the interior is a slow process (Rao and Shashidhar, 2016).

## 3 Sampling

This study has been conducted on the samples of the KBH1 borehole which was drilled near Rasati (17°22′38.5″N,

73°44′27.8″E, about 4.7 km from the Koyna Dam). The borehole KBH1 has passed through the Ambanenli and Poladpur Formation of Wai Subgroup in the Deccan Trap Basalt upto 932.5 m and then extended into the basement rocks underlying the Deccan Trap Basalt upto 1522 m depth (Sinha et al., 2017) **(Fig. 2)**. The basement rocks here are composed of granite, granite-gneiss, and migmatite gneiss, overall granitoids in nature. (Misra et al., 2017b; Piyal et al., 2021; Shukla et al., 2022). Quartz and feldspars are the primary felsic constituent minerals, whereas

biotite, pyroxene and hornblende are the mafic components found in the basement rocks. Within the recovered core samples of KBH1, several networks of interconnecting fractures filled up with secondary minerals and numerous small-scale faults have been seen in two distinct depth sections, 940-1075 m and 1142-1238 m. (Misra et al., 2017b). Fissures (Mode-I) as well as Shear Fractures both have been found with fracture angles 30°-40° and 10°-20° with respect to the borehole axis and reveal displacement of pre-existing fabric indicating that they

were probably generated due to the stress build-up caused by the impoundment of the dams.

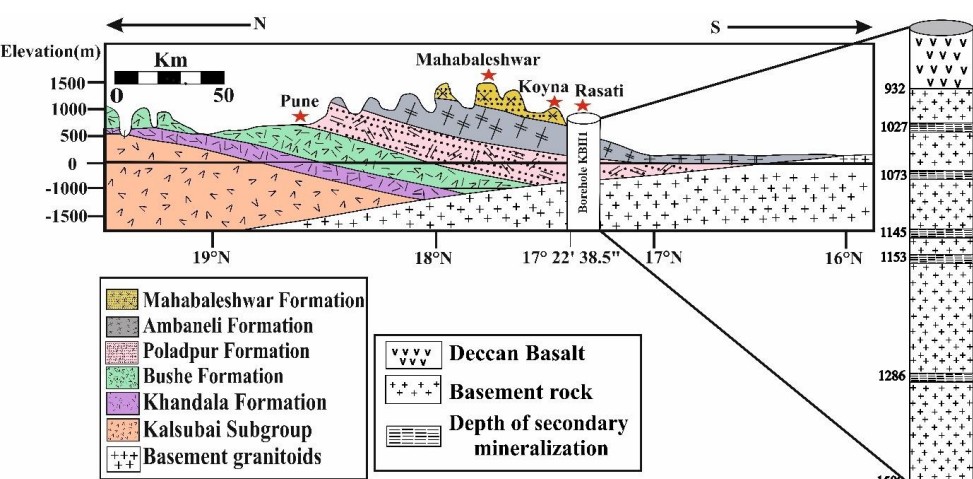

**Figure 2.** Schematic diagram along Western Ghat escarpment between 19°00' N to 16°00' N, showing the distribution of several chemolithostratigraphic formations at and around the studied borehole (KBH1) (modified after Sinha et. al., 2017). The litholog of the studied borehole exhibits the distribution of the porous and vesicular

Deccan basalt underlain by the granitoid basement rocks in which depths of alteration or secondary mineralisation have been marked.



In our study, the samples have been collected from 14 different depths throughout the entire borehole ranging from 932.5 m (starting of the granitoid basement) to 1522 m (end of the KBH1 borehole). The altered samples were collected through mesoscopic observations. The altered samples recovered from borehole KBH1 show the presence of a number of secondary minerals, including chlorite, epidote and others, as well as the precipitation of carbonate at various depths, particularly along the fractures and slip surfaces of the small-scale faults. The fluid-induced alteration and subsequent precipitation in the granitoid basement are indicated by the mineralized network of fractures, predominant calcification on the slip surface, and significant greenish tint or precipitation of green-coloured minerals (chlorite and/or epidote) **(Fig. 3a-c)**. Additionally, the granitoid's soil-like friability at various depths certainly demonstrates the strong interaction of the fluid with the rock **(Fig. 3d).**

Altered samples (KBH1_378, KBH1_379, KBH1_380, KBH1_381, KBH1_382, KBH1_383) are mainly found within two distinct depths (940-1075 m and 1142-1238) m at which the brittle deformation features like faults and fractures are encountered as mentioned earlier, whereas the rest samples below 1286 m depth devoid of any such features except only a few isolated fractures, which is why fluid percolation was hardly tenable in this depth zones making the basement rocks relatively fresh and unaltered.

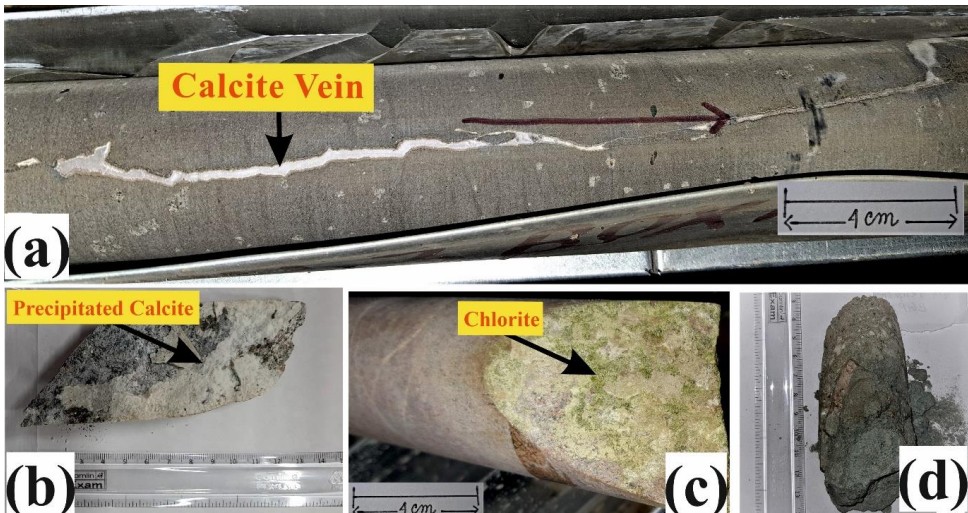

**Figure 3. Mesoscopic observations on the core samples of the borehole KBH1.** (a) Vein filled up with white-coloured minerals indicating fracture mineralisation during fluid percolation along fracture; (b) Calcite precipitated on the slip surface; (c) Greenish tint on the slip surface indicating the presence of secondary chlorite or epidote; and (d) Friable nature of samples representing the dissolution caused by fluid-rock interaction.



## 4 Methods

To identify the neoformed secondary minerals and comprehend the mechanism of fluid-rock interaction, core samples collected from the altered and modified zones of the KBH1 borehole have been studied using four different analytical tools.

I. With the help of Leica DM2700 P and Leica DM750 P, optical microscopic analyses of the petrographic thin sections made from the altered and unaltered samples have been carried out for mineralogical characterization.

The occurrences of secondary mineral growth over the primary minerals, grain boundary alteration, and minerals formed along the network of fractures were given special attention.

II. Field Emission Scanning Electron Microscopy (Model- JEOL JSM 7610f) has been conducted on the altered rock pieces and the mesoscopically discernible secondary precipitations extracted from the fracture zones by the Dremel Microdrill machine (Dremel 8220-1/28). On the same altered rock pieces, Energy Dispersive

Spectroscopic (EDS) analysis in three different modes (i.e., point, line, and area) has also been performed. Additionally, secondary minerals like chlorite, calcite, and mineralized veins identified in the thin sections have been reanalysed under FESEM-EDS for more detailed characterization and an approximate estimation of the elemental composition that has helped to provide an overview of the mineral transformation process.

III. Bulk mineralogical powder X-ray diffraction (pXRD) analysis (model- X'PERT[3] powder; PANalytical) was

conducted to determine the most prevalent primary and secondary minerals present in the samples selected based on the outcome of the FESEM-EDS studies.

Additionally, oriented clay slides have undergone clay mineralogical XRD examinations to validate the presence of clay minerals. However, unlike sediments or soils, it is more difficult to separate clay from samples of hard rock. Hence, the extremely altered samples, in which the presence of clay minerals has been confirmed by the

SEM-EDS and bulk XRD studies, have been gently crushed with the pestle in an agate mortar rather than grinding, which might give rise to clay-sized primary minerals. These crushed materials were run through 230 mesh-sized sieves every 30 minutes to separate the <5-micron particles from the comparatively larger-sized particles. These sieved residues were then subjected to ultrasonic disaggregation for <30 minutes, to prevent the flocculation of clay-sized particles, which were then separated using the Stokes' principle. The collected clay

fractions of each sample were separated into three parts and finally mounted on three different glass slides. In order to distinguish between distinct clay mineral species (1:1 or 2:1 type), one slide from each sample was run on XRD in air-dried condition whereas, the second one was run after ethylene glycol saturation and the last one was run after heating at 550°C. However, due to the extremely limited amount of samples, it was not possible to extract enough clay minerals from each and every sample, making it impossible to run clay mineralogical

XRD on every sample. However, sufficient clay fraction recovery from a series of samples was made to allow generalization and avoid limiting the scope of the study.

IV. The samples taken from 936 m depth to 1521 m depth of the borehole KBH1 have been analysed using Inductively Coupled Plasma Optical Emission Spectroscopy (Model: Agilent 5800 ICP-OES) in order to understand the reaction kinetics Disk mill was used to grind the samples to around, and the agate mortar was

used to homogenise them for equal mixing. -230 mesh size powdered samples were transferred into PTFE bombs to a dose of around 30 mg for open-vessel acid digestion using $HF/HNO_3/HClO_4$ (ultrapure) acids mixture. The samples were initially, digested using an acid mixture of $HNO_3:HF:HClO_4$ in a 1:2:1 ratio at 150 °C for an



overnight period with their lids on. The digested samples were completely dried before being cooked once more for 12 hours using $HNO_3$:HF:$HClO_4$ in 2:1:1 ratio. In order to get the elemental concentrations (ppm), which

were afterwards converted into oxide concentration (wt.%) for a simpler and better comprehension, the thoroughly dried digested samples were diluted into 3% $HNO_3$ solution (Dilution Factor 5555.567). The method of sample preparation and analyses is validated using certified Reference Materials (RGM-2 and BHVO-2). The relative standard deviation (r.s.d.) for the key study variables is less than 1% for each analyte element. The measurement values of each Certified Reference Material are within the certified values (<5% difference).

## 5 Results

### 5.1 Optical Microscopy

Petrographic thin sections made from the different depths of the granitoid basement rocks, comprising plagioclase, quartz, and biotite as the primary minerals, underwent optical microscopic investigation. The alteration in the samples at shallow depth (KBH1_378, KBH1_379, KBH1_381, KBH1_382) is characterized by the presence of

chlorite, epidote, and calcite (**Fig. 4a, b**). However, at relatively deeper levels (KBH1_383), it is manifested only by the calcite and dissolved plagioclases (**Fig. 4c**), even if chlorite is present that too in small quantity and restricted to only a small part of the thin sections. The biotitic remnants over the neoformed chlorite state about the transformation of chlorite into biotite (**Fig. 4d**), whereas the dissolved plagioclase surfaces suggest that the dissolution of plagioclase and the chloritization of biotite are somehow correlated. Albites are also found along

with chlorite as brown-coloured grains resembling biotite but the 9°-10° extinction angle under cross-polarized light (XPL) and Elemental Dispersive Spectroscopy (EDS) confirm them as albite (**Fig. 4e**). This brownish hue appearance may be a sign of surface dissolution because of the intensive fluid interaction. Numerous fractures filled up with calcite and a few chloritic veins indicate the secondary origin of these minerals and reveal that precipitation has occurred due to fluid flow along the earlier formed interconnected fractures (**Fig. 4f, g**).

In addition, the chess-board pattern extinction in the quartz grains suggests high strain rates when prismatic glide occurs along with the basal glide and the marker offset within the plagioclase grains reaffirms the brittle deformation as a result of which fractures were formed (**Fig. 4h, i**). Chloritic grains dissected by epidote have been found in a few altered samples denoting the formation of epidotes after the genesis of chlorite (**Fig. 4j**). This cross-cutting relationship helps to interpret the sequence of events, i.e., there were numerous stages of alteration

caused by the interaction of the fluid with the rock bodies and formation of epidote may be dependent on the elements remobilized during chloritization of biotite (**See discussion**). Furthermore, the flow-like appearance of chlorite and kinking within the chlorite grain in the altered sample at greater depth (KBH1_381) confirms the formation of chlorite as the result of fluid-rock interaction and strain accommodation due to continuous stress buildup (**Fig. 4k**). Kinking occurs in chlorite crystal in response to the c-axis parallel strain when slip on a single

plane cannot facilitate homogeneous deformation, so it bends through shortening parallel to basal planes. Kinking may have larger implications in seismicity which will be discussed in the later part.



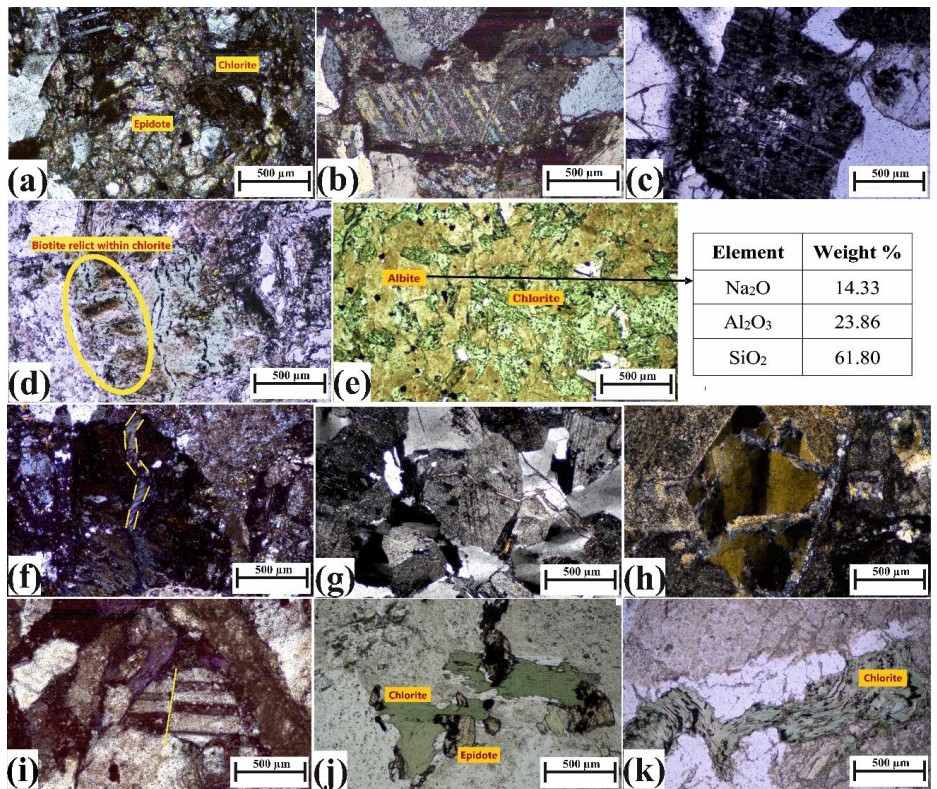

**Figure 4.** Photomicrographs of the altered samples collected from different depths of the KBH1 borehole. (a) The

assemblage of chlorite and epidote found under Cross-Polarised light (XPL); (b) Calcite with its characteristic polysynthetic twinning observed under XPL in the sample collected from the fault slip surface. (c) Dissolved plagioclase surface noticed in the sample KBH1_383; (d) Biotite remnant found in neoformed chlorite under PPL demonstrating its biotitic origin; (e) The association of chlorite with the brownish albite resembling biotite found under Plane Polarised Light (PPL). Corresponding EDS data corresponds to its albite-like sodic composition; (f)

Fractures filled up with chlorite under XPL represent fractures as the pathway of fluid percolation and post-deformation fluid-rock interaction.; (g) Precipitation of calcite within fractures cutting Plagioclase crystal as the evidence of dissolution-precipitation mechanism during fluid-rock interaction; (h) The chess-board pattern extinction suggests high strain rates corresponding deformation events; (i) The marker offsets within the plagioclase grain under Cross-polarized Light (XPL) denotes brittle deformation as a result of which fractures

were developed; (j) Cross-cutting relationship of chlorite and epidotes reflects that epidote was formed after chlorite; (k) The flow-like appearance of chlorite and kinking within the grain observed under Plane-Polarized light (PPL) confirms the formation of chlorite as the result of fluid-rock interaction and strain accommodation. Kinking represents the shortening parallel to basal planes in response to the c-axis parallel strain and indicates the continuous stress build-up in the host rock.




### 5.2 FESEM-EDS investigations

FESEM image of the thin section of KBH1_378 shows surface leaching of the host minerals and the corresponding EDS data reveals the absence of $K_2O$ in the residual mineral surface **(Fig. 5)**. This is probably an indicator of the removal of the potassium interlayer sheet during the transition of the biotite into chlorite. On the other hand, thin-section scanning of KBH1_379 and KBH1_381 under FESEM and EDS identifies the precipitation of calcite along fractures confirming the petrographic observations under the optical microscope **(Fig. 6a and c)**. However, EDS investigation of the mineralized vein present in the KBH1_379 reflects the elemental composition characteristic of chlorite and calcite **(Fig. 6b)**. All these findings indicate that in addition to precipitating calcite, percolated fluid along the fractures also formed the chlorite. Biotite chloritization and calcite precipitation may be connected which needs to be addressed by reaction mechanics.

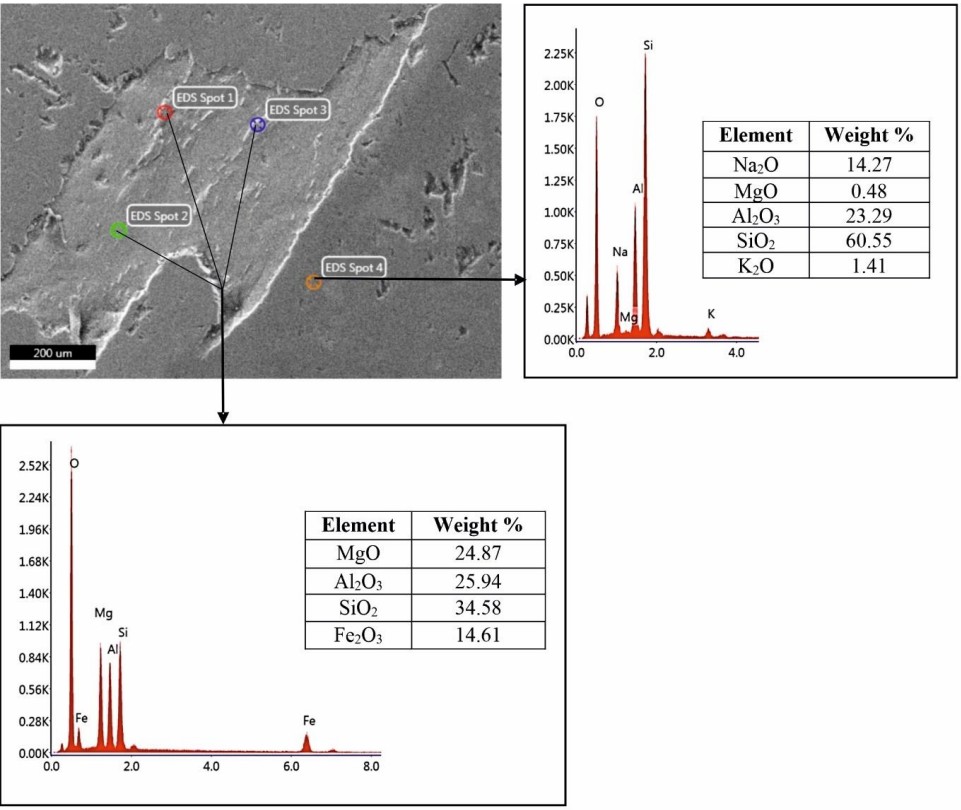

**Figure 5.** FESEM image and the corresponding EDS data of the thin section prepared from KBH1_378 (1027 m depth). The image clearly shows the leaching of the host mineral surface which is further supported by the EDS data. The EDS data of the three spots 1, 2, and 3 within the relict surface shows the absence of $K_2O$ although spot 4 within the surrounding surface has little presence of this major oxide. This is probably a signature of the removal of the potassium interlayer sheet during the transition from biotite to chlorite.







(a)

(b)

(c)



**Figure 6.** FESEM images and the corresponding EDS data of a few petrographic thin sections of the altered
samples, reveal evidence of chemical and subsequent mineralogical transformation. (a) FESEM image of a part
of the thin section of KBH1_379 (1073 m depth) shows the precipitation of a mineral, the texture of which is quite
different from that of the host minerals, whereas EDS data in point and areal scanning mode reveals that guest
mineral as Calcite; (b) FESEM image of a mineralised vein in the same section in which EDS data of the vein
minerals reveals the elemental composition characteristic of chlorite and calcite representing the fracture-scale
fluid-rock interaction and subsequent mineralisation; (c) FESEM image of a wide channel fracture in the thin
section of the sample KBH1_381 (1153 m depth) and the corresponding EDS data of the fracture filling material
reflects the composition of calcite.

**5.3 XRD analysis**

The samples collected from 1027 m depth (Sample ID- KBH1_378), show the presence of albite and clinochlore
in the bulk mineralogical XRD, whereas the samples recovered from 1073 m and 1153 m (Sample ID- KBH1_379
and KBH1_381) show the presence of calcite in addition to clinochlore and albite **(Fig. 7a, c, and f)**. However,
samples taken from 1145 m below the surface reveal the presence of illite **(Fig. 7e)**. The presence of clay minerals
like chlorite, illite, etc. in the same set of samples are also confirmed by clay mineralogical XRD **(Fig. 7b and d)**.







**Figure 7.** XRD data of the KBH1 borehole. (a) Bulk Mineralogical XRD of the sample collected from 1027 m

depth shows the occurrences of clinochlore and albite; (b) Clay Mineralogical XRD of the sample (Air-dried,



glycolated and heat treated at 550 C) collected from 1027 m depth confirms the presence of chlorite; (c) XRD of the bulk sample collected from 1073 m depth shows the occurrences of clinochlore, albite and calcite; (d) XRD of the clay separated from the sample of 1073 m depth, run in three different modes (Air-dried, glycolated and heat treated at 550 C), reaffirms the presence of chlorite; (e) Bulk Mineralogical XRD of the sample collected

from 1145 m depth represents the occurrences of clinochlore, albite and calcite; (f) Bulk Mineralogical XRD of the sample collected from 1153 m depth exhibits the presence of clinochlore, quartz, albite and calcite.

**5.4 ICP-OES analysis**

The concentration of a few major oxides shows significant variation in the ICP-OES analysis (see **Table 1**). As shown in **Fig. 8**, samples taken from the depths 1027 m, 1073 m, 1145 m, 1153m and 1286 m (Sample Id-

KBH1_378, KBH1_379, KBH1_380, KBH1_381, KBH1_382, KBH1_383, KBH1_390) exhibit substantial variation in the concentration of major oxides, such as- MgO, FeO$^t$, CaO, TiO$_2$, and K$_2$O (in wt. %). The samples KBH1_378, KBH1_379, KBH1_380, KBH1_381, and KBH1_382, show a significant increase in the concentration of MgO and FeO$^t$ and relatively little enrichment of TiO$_2$. However, K$_2$O exhibits exactly opposite behaviour in the same set of samples.  The increasing trend of MgO, FeO$^t$, and TiO$_2$ indicates the incorporation

of those oxide-forming major elements in the crystal lattice of the constituent minerals reflecting the formation of chlorite, whereas the decreasing trend of K$_2$O represents the removal of K from those samples mainly supporting the dissolution of biotite. Similarly, the increase in CaO in KBH1_378, KBH1_379, KBH1_381, and KBH1_383 represents the precipitation or formation of Ca-bearing minerals like calcite and epidote, indicating that their formation is not distinct geochemical phenomena rather they are somehow related to one another (see more details

in the discussion section).

Furthermore, the samples KBH1_380 and KBH1_390 exhibit a rising trend of K$_2$O and likely signify the formation of illite where the K has come from the previously dissolved biotite. Thus, major oxide data of the altered samples in comparison to the unaltered samples (KBH1_384-389, 391-394) reaffirms the formation of chlorite, epidote, and illite along with the precipitation of calcite confirming the dissolution of biotite as evidenced

in the optical microscopy and XRD investigations. Later parts explain the method by which new minerals arise as a result of fluid-rock interaction at the expense of a few other existing mineral phases.



| | Depth | Al$_2$O$_3$ | CaO | FeO$^t$ | K$_2$O | MgO | Na$_2$O | TiO$_2$ | LOI | Total |
|---|---|---|---|---|---|---|---|---|---|---|
| KBH1-377 | 936.76 | 14.81 | 0.81 | 1.49 | 4.07 | 1.35 | 4.41 | 0.17 | 1.68 | 28.79 |
| KBH1-378 | 1027.81 | 21.34 | 6.06 | 9.24 | 1.31 | 5.15 | 5.56 | 1.46 | 4.96 | 55.08 |
| KBH1-379 | 1073.75 | 16.40 | 4.47 | 8.08 | 0.58 | 8.83 | 4.03 | 0.90 | 10.08 | 53.37 |
| KBH1-380 | 1145.61 | 13.47 | 1.15 | 3.77 | 3.19 | 3.94 | 2.38 | 0.32 | 6.08 | 34.3 |
| KBH1-381 | 1153.78 | 15.12 | 3.74 | 4.05 | 1.18 | 2.05 | 4.25 | 0.55 | 4.47 | 35.41 |
| KBH1-382 | 1201.3 | 14.48 | 0.43 | 4.53 | 0.19 | 4.70 | 5.47 | 0.88 | 3.99 | 34.67 |
| KBH1-383 | 1286.04 | 14.02 | 3.19 | 1.25 | 1.19 | 0.58 | 3.59 | 0.15 | 4.05 | 28.02 |
| KBH1-384 | 1300.48 | 16.55 | 2.52 | 4.46 | 1.73 | 2.08 | 5.12 | 0.54 | 1.75 | 34.75 |
| KBH1-387 | 1360.07 | 14.56 | 1.36 | 0.50 | 1.22 | 0.16 | 8.80 | 0.06 | 1.05 | 27.71 |
| KBH1-388 | 1379.81 | 13.08 | 2.26 | 3.01 | 1.26 | 1.51 | 3.98 | 0.43 | 1.33 | 26.86 |
| KBH1-389 | 1415.65 | 15.04 | 2.90 | 0.72 | 1.51 | 0.27 | 5.79 | 0.10 | 0.76 | 27.09 |
| KBH1-390 | 1432.97 | 16.04 | 2.10 | 2.11 | 5.30 | 0.48 | 4.00 | 0.22 | 1.12 | 31.37 |
| KBH1-391 | 1447.63 | 15.27 | 2.28 | 4.38 | 3.94 | 1.35 | 4.59 | 0.49 | 0.38 | 32.68 |
| KBH1-392 | 1455.7 | 14.22 | 2.14 | 4.20 | 3.80 | 1.28 | 4.38 | 0.46 | 1.48 | 31.96 |
| KBH1-393 | 1472.01 | 15.16 | 3.78 | 6.71 | 1.77 | 2.34 | 4.31 | 1.02 | 1.23 | 36.32 |
| KBH1-394 | 1480.13 | 17.35 | 3.47 | 2.41 | 1.43 | 0.73 | 5.35 | 0.30 | 1.23 | 32.27 |

**Table 1.** The concentration of a few major oxides (in wt. %) in the samples collected upto 1500 m depth from the borehole KBH1.



**Figure 8.** Depth-wise variation profile of a few major oxides. Depth-wise variation in the concentration of major oxides reveals significant disparity at the depths which are marked by neomineralisation and/or dissolution of certain mineral phases.





**6 Discussion**

The mineralogical and supporting geochemical investigations have shown that fluid migration through faults and fractures and subsequent fluid-rock interaction in the basement rocks of the Koyna-Warna Seismogenic Region have caused chemical changes that have led to the formation of new secondary minerals like chlorite, illite, epidote, and calcite. The findings are not homogeneous in all the altered samples as also found in XRD; most of the altered samples are found as highly chloritized along with calcite and epidote (KBH1_378, KBH1_379,

KBH1_382, KBH1_381), a few of them are showing only the dominance of calcite (and KBH1_383) and rest others are characterized by the presence of illite (KBH1_380 and KBH1_390). However, this entire assemblage of neoformed minerals may be the products of a similar kind of alteration which needs to be understood in terms of the gain and loss of the major oxides found in geochemical investigation. Hence, the results of this study are significant in determining the transformation pathway from parent minerals to the newly formed secondary

minerals, which is ultimately helpful in interpreting the nature and extent of subsurface fluid-rock interaction even though the source of the fluid cannot currently be inferred without more thorough investigation.

**6.1 Mass balance calculation & Isocon plot**

To determine the relative losses, gains or mobility of elements in the altered samples, the concentration of major oxides has been subjected to mass balance calculation **(Table S1)**. The outcome of the mass balance calculation

has been represented graphically, in which concentrations of elements in the altered rocks ($C_a$) are plotted against the average concentration of those in the relatively fresh/unaltered rocks ($C_o$: KBH1_384-389, 391-394) **(Fig. 9)**. Two isocons have been drawn in case of every altered sample; one isocon has been determined as the best-fit line by graphically searching the elements that lie on or near a line passing through the origin (named as Best-fit isocon) (Grant, 1986), another has been extended by clustering and selecting the elements that show minimum

gain/loss (here $Al_2O_3$) with respect to other elements (named as Immobility Isocon). However, the outputs are almost the same in both cases. For a better representation of the mass gain/loss and to avoid congestion, the data have been scaled to even values, in decreasing concentrations following Grant (2005) **(Table S2)**.

Mass Balance calculation and isocon plotting of the altered samples (Fig. 9a, b, d, e) indicate the loss of $K_2O$ and simultaneous gain of MgO, and FeO, confirming the dissolution of biotite and neoformation of Mg-Fe chlorite,

as found in the optical microscopic investigation (Fig. 4d). Additionally, the gain of CaO in the samples (Fig. 9a, b, d, f) denotes the precipitation of calcite, most of which appeared as several veins filling materials under the microscope (Fig. 4g). However, its loss in the sample in KBH1_380 along with the contrasting gain of $K_2O$ designates the dissolution of plagioclase and formation of illite respectively (Fig. 9c, e, g).

Apart from these major oxides, LOI is another important component which includes various volatile matters such

as free water, carbonate, etc. So, its gain in all the altered samples indicates the noteworthy fluid-rock interaction and subsequent formation of hydrophilic clay minerals as well as carbonate precipitation, the mechanism of which will be discussed in the later section in terms of mass gain and loss of the above-mentioned major oxides.





**(a)**      **(b)**      **(c)**

**(d)**      **(e)**      **(f)**

**(g)**

**(h)**      **(i)**      **(j)**

**(k)**      **(l)**      **(m)**

**(n)**



**Figure 9.** Graphical representation of the mass balance calculation of the altered samples. (a-g) Isocon diagrams
showing relative loss and gain of chemical constituents in the altered samples with reference to the average
concentration of the relatively unaltered samples ($C_O$). Actual concentrations are scaled to even values, in
decreasing concentrations following Grant, 2005 (See table S2). Elements plotted above isocon lines indicate
their gain whereas the lost elements are plotted below this line; (h-n) Bar diagrams exhibiting the loss and gain of
elements with respect to the average concentration of the relatively unaltered samples ($C_O$).

**6.2 Mechanism of secondary mineralization**

Members of the chlorite mineral group typically form at low-temperature through the Dissolution-
Recrystallization (DR) mechanism, from certain minerals such as biotite, pyroxene, amphibole, olivine, etc. In
our study, biotite partially replaced by chlorite or chloritic growth over biotite planes indicates the biotitic origin
of chlorite. Nishimoto and Yoshida (2010) claimed, that when the fluid/rock ratio is low because of the
comparatively low porosity and low fracture density, the biotite chloritization can be a signature of the early stages
of the successive hydrothermal alteration. In general, the transformation of biotite into chlorites under low-grade
metamorphic conditions occurs through the following reactions (Chayes, 1955):

Biotite + quartz + $H_2O$ → chlorite + orthoclase

However, the percolating fluid in hydrothermal settings may not be isochemical instead can contain several
additional components. Chloritization therefore happens in conjunction with other secondary phases including
clay minerals, calcite, epidotes, Titanite etc. (Schwartz, 1958; Parneix et al., 1985)

Veblen and Ferry (1983) postulated two distinct methods for the conversion of biotite into chlorite, which are
discussed below-

**(I) One biotite to one chlorite (1Bt→1Chl)-**

$KMg_3[AlSi_3O_{10}] (OH)_2 + 3H_2O + 1.5O_2 + 2Mg^{2+} + Al^{3+} →[Mg_2Al] (OH)_6Mg_3[AlSi_3O] (OH)_2 + K^+$

            (1 biotite)                                   (1 chlorite)

For convenience, in the current article, this is referred to as the 1B1C mechanism, which is caused by the fluids,
already enriched in $Mg^{2+}$ and $Al^{3+}$. One molecule of chlorite is produced at the expense of one molecule of biotite
by the brucitisation of the potassium interlayer following the increase in volume. The space created by
fracture/crack formation or displacement due to fault generation further may facilitate this increase in volume.

**(II) Two biotite to one chlorite (2Bt→1Chl)-**

$2KMg_3[AlSi_3O_{10}] (OH)_2 + 5H_2O + 0.5O_2 → [Mg_2Al](OH)_6Mg_3[AlSi_3O_{1O}](OH)_2 + 2K^+ + Mg^{2+} + 3H_2SiO_4$

            (2 biotite)                         (1 chlorite)

This reaction mechanism (2B1C) differs significantly from the 1B1C mechanism in that no dissolved components
are present in the reacting fluid, and the brucitisation of the talc-like layer results in the production of one molecule
of chlorite at the expense of two molecules of biotite. This occurs due to the inheritance of an octahedral sheet
together with the loss of an adjacent tetrahedral sheet and a potassium interlayer sheet in one biotite layer (Veblen
and Ferry, 1983). This reaction demonstrates that the single chloritization from biotite can occur without any sort
of external chemical components except water and $H^+$. Many tetrahedral components, including silicon and



interlayer potassium from biotite, are released together with chlorite. Additionally, the octahedral components are dissolved in the fluid and can facilitate further mineral alteration. Elements like titanium, aluminium, iron, magnesium etc. may also be released in this mechanism with a significant reduction in volume due to the vacancy and disarray of sheets in the crystal structure of biotite (Eggleton and Banfield, 1985).

        In our study, the mass balance calculation of a few major oxides shows the significant gain of MgO & FeO$^t$ in
most of the altered samples and the loss of $K_2O$ in the same set, suggesting that chloritization may be a combination of both mechanisms in such a ratio whereby the volume change is very small as reported by Ferrow and Ripa (1991), and Yau et al. (1984). In particular, mass loss of $K_2O$ represents the dissolution of the potassium interlayer sheet due to fluid activity whereas mass gain of MgO can be attributed to the replacement of that vacant position by a brucite-like sheet which acts as an octahedral sheet and preserves the spatial geometry of the biotite
crystal (e.g., vacancy and disarray of sheet). Thus, the 2B1C formation mechanism is followed by the 1B1C mechanism.

        It is also interesting to note that the gain of FeO is relatively less than that of MgO, which explains why the bulk mineralogical XRD significantly reveals the presence of clinochlore (Mg-rich) rather than chamosite (Fe-rich). However, the EDS analysis of biotite and chlorite shows, that the Mg content of the latter is higher than that of
the former (Fig 5). This additional amount of Mg may be brought about in the system by the percolating fluid while flowing through the Mg-rich part of the basaltic top containing magnesian clinopyroxene and calcic plagioclase phenocrysts (Banerjee and Mondal, 2021). Thus, the role of the fluid can also be confirmed in the alteration of mineral assemblage.

        Nevertheless, the KBH1_383, in which chlorite has not been observed, revealed the loss of both MgO and FeO
and such complete absence of chlorite at relatively greater depth may be due to the diminished activity of the reacting fluid.

        Although the immobility isocon has been plotted through $Al_2O_3$ due to its relatively immobile behaviour of Al as found in bulk geochemical analyses still the EDS analyses of chlorite reveal the content of $Al_2O_3$ little higher (25.94 wt. %) than that of parent biotite (23.25 wt. %) (Fig. 5). Actually, Al is not completely an inert component,
it can migrate along small distances from one mineral to another mineral (Parneix et al., 1985). So, in the case of biotite chloritization, the Al can be transported from plagioclase during its dissolution and subsequent alteration, thus conserving the total mass of $Al_2O_3$.

        Plagioclase + $Fe^{2+}$ + $Mg^{2+}$ + $K^+$+ $H_2O$→ albite + illite + $Ca^{2+}$ +$Al^{3+}$                    (Parneix et al., 1985)

        This reaction explains why the majority of plagioclases in altered samples have been found partially dissolved
(vugs are filled up by chlorite or carbonate), whereas the residue mass of plagioclase displays the albitic nature. Such partial dissolution of plagioclase advocates the little loss of $Na_2O$ as observed in the mass balance calculation and isocon plots. Moreover, during this process along with $Al^{3+}$, $Ca^{2+}$ is also released which facilitates the precipitation of the carbonate minerals like calcite and elucidates the gain of CaO in the altered samples. Similarly, the formation of illite through this reaction justifies the gain of $K_2O$ in KBH1_380. On the other hand, the Al
released here not only aids in the formation of chlorite but also the production of epidote, However, because of a reduction in volume following the synthesis of chlorite, the other minerals formed during biotite chloritization rely on the remaining vacancies.



1 biotite + $Al^{3+}$ + $Mg^{2+}$ + $Fe^{3+}$ + $Ca^{2+}$ + $H_2O$ + $H^+$→ chlorite + epidote + $TiO_2$ + $H_4SiO_4$ + $K^+$ (Parneix et al., 1985)

This reaction accounts for the very little gain of $TiO_2$ as evidenced in the mass balance calculation and subsequent isocon plots.

This is how the fluid flow along the pre-existing fractures and its subsequent interaction with the basement rocks of the Koyna-Warna Seismogenic Region leads to biotite chloritization with significant volume reduction. This volume reduction may induce fractures that permeate the biotite crystal and form an anastomosing network of fluid-filled fractures, as also experimentally shown by Janssen et al. (2018) during the transformation of ilmenite to rutile. Consequently, the fluid flow gets increased and promotes plagioclase dissolution which facilitates the micropore formation and aids in the mobility of the elements via the hydrothermal fluid to advance the biotite chloritization. Thus, biotite chloritization influences plagioclase dissolution which provides the pathway of chloritization making a cyclic loop (Yuguchi et al., 2019). But this is not an infinite cycle, the inflow of $H^+$ during biotite chloritization increases the pH of fluid which promotes the plagioclase dissolution. With the advancement of plagioclase dissolution, the pH of the fluid gets decreased, so the entire cycle may be ceased when the gradually decreasing pH touches the threshold limit and plagioclase dissolution gets stopped (Yuguchi et al., 2019). This sort of fluctuation of pH due to the inflow of $H^+$ during biotite chloritization and subsequent plagioclase dissolution may be identified as a reason why the indicative pH of such an alteration process ranges from 7 to 5 (Fulignati, 2020). Although the neoformed mineral assemblage, chlorite + illite + epidote + albite, are the characteristic of greenschist facies of metamorphism, the presence of fractures filled up with chlorite or chloritized veins (Fig. 4f) and occurrence of calcite (Fig. 4b and 4g) along with the other secondary minerals probably represents propylitic grade of hydrothermal alteration involving carbonatisation in association with hydration at moderate temperatures (Mathieu, 2018). Notwithstanding it is very difficult to comment on the exact temperature of such subsurface hydrothermal alteration without more detailed investigations, the presence of illite and chlorite indicates a temperature of above 200–220° C up to about 350° C (Fulignati, 2020). However, the studies based on the heat flow, thermal conductivity and radiogenic heat production data suggest temperatures in the range of 130–150 °C at a depth of 6 km in this region so at the depth of our investigation the temperatures should be less than 150 °C (Gupta et al., 2015). Therefore, this remaining heat budget may be accounted for by the heat generation through friction and fracturing within the fault slip zone or by the flow of hydrothermal fluid ascending upwards through the interconnected fractures.

**6.3 Quantification of alteration**

**6.3.1 Ishikawa Alteration Index**

The enrichment and depletion of major elements due to the fluid-rock interaction have given rise to different degrees of hydrothermal alteration at different depths. Hence, several Alteration Indices have been used to estimate the degree of alteration in terms of different sets of mobile and immobile components. The Alteration Index (AI) proposed by Ishikawa et al. (1976) has been used here as a measure of chloritization, which is defined as,

$$AI = 100 \times (K_2O + MgO) / [K_2O + MgO + Na_2O + CaO]$$





This index ranges from 20 to ~60 for unaltered or fresh rocks and between 50 and 100 for hydrothermally altered
rocks, whereas an AI value of 100 indicates completely altered rock containing chlorite (Large et al., 2001).

Graphical representation of the Ishikawa Alteration Index with increasing depth has shown a relatively higher
degree of alteration in the samples at shallow depth in comparison to the unaltered/fresh samples found at a deeper
level within the Koyna-Warna basement granitoids **(Fig. 10a)**. The gain of MgO during chloritization has resulted
in such an increased degree of alteration in the samples KBH1_377, KBH1_378, and KBH1_379, whereas the
highest Alteration Index found in KBH1_380 is the expression of excessive gain of $K_2O$ and formation of illite.
The samples KBH1_381 and KBH1_383 have portrayed low to moderate Alteration Index, probably due to the
extreme calcification along the fractures and fault slip surfaces.

### 6.3.2 Chlorite-Carbonate-Pyrite Index (CCPI)

During biotite chloritization the concentration of FeO has also fluctuated as shown in Fig. 8. Hence, it is equally
important to consider the variation of FeO along with the variation of other major oxides like MgO, $K_2O$, $Na_2O$,
and CaO. Therefore, the Chlorite-Carbonate-Pyrite Index (CCPI), introduced by Large et al. (2001), has also been
used here. CCPI is defined as,

$$CCPI = 100 \times (FeO + MgO) / [K_2O + MgO + Na_2O + FeO]$$

Graphical representation of CCPI along increasing depth shows the highest degree of alteration in KBH1_379
rather than KBH1_380, which was found in the Ishikawa Alteration Index **(Fig. 10b)**. A possible explanation
behind such change is that CCPI is more prominent for chloritization which is undoubtedly highest in the sample
KBH1_379, as found in the mass balance calculation. Rest observations are almost consistent with those in the
Ishikawa Alteration Index.

### 6.3.3 Alteration Box Plot

The Alteration Box Plot, proposed by Large et al. (2001), has also been used here for a more accurate measurement
of the degree of chloritization. It combines the Ishikawa Alteration Index along the X-axis and the CCPI along
the Y-axis **(Fig. 10c)**. The minerals like calcite, albite, epidote, etc. are plotted on the left-hand CCPI axis or the
lower AI axis, and hydrothermal mineral such as chlorite is plotted on the right-hand CCPI axis. Thus, a diagonal
line joining epidote to K feldspar effectively separates the box plot into two segments in which the upper half
represents the hydrothermal alteration and the trend towards the upper right corner signifies the increasing degree
of such alteration, more specifically chloritization. The plotting of all the altered samples significantly falls in the
field of hydrothermal alteration and KBH1_379 shows the highest degree of alteration as also found in CCPI.
Calcification in KBH1_381 and KBH1_383 are also clearly discernible from their trends towards carbonisation
in this plot.



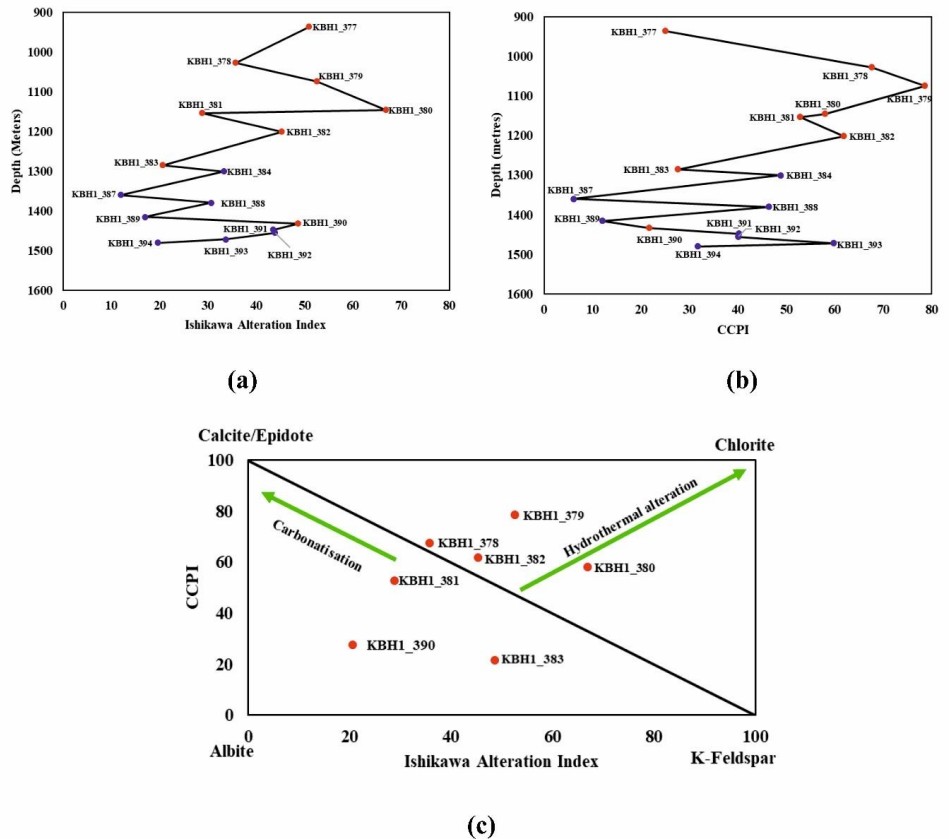

**Figure 10.** Graphical representation of the Ishikawa Alteration Index, Chlorite-Carbonate-Pyrite Index (CCPI) and Alteration Box Plot. (a) Plotting of Ishikawa Alteration Index with increasing depth shows its higher degree in the samples at shallow depth; KBH1_377, KBH1_378, and KBH1_379 are showing relatively higher values of AI, the highest Alteration Index (AI) is observed in KBH1_380 whereas KBH1_381 and KBH1_383 are showing low to moderate Alteration Index. (b) CCP Index of the samples shows a similar trend as the Ishikawa Alteration Index except the highest degree is encountered in the sample KBH1_379, as CCPI accounts for FeO concentration representing a more reliable indicator for chloritization. (c) Alteration box plot of the altered samples, in which KBH1_378, KBH1_379, KBH1_380, KBH1_382 are falling in the hydrothermal field whereas, KBH1_381, KBH1_383 are trending towards the carbonatisation.

**6.4 Implications of the secondary mineralization**

The Koyna-Warna Seismogenic region has undergone three phases of deformation, according to a recent study by Misra et al. (2022), the first of which is represented by the development of gneissosity in the granitic rocks, the second by shearing events that caused pre-existing fabrics (S1) to be overprinted and transposed by the high strain S2 fabrics, and the third by the emergence of anastomosing fractures as a function of brittle deformation such as faulting. The fractures cutting the plagioclase grains and fillings of chlorite and calcite throughout the network of fractures, as documented in our study (Fig. 4f, g), indicate that the percolation of fluid and subsequent



formation or precipitation of secondary minerals along the fractures took place after the third deformation event in which faulting occurred and fractures were developed. Unfortunately, the information available till now is not adequate to determine whether the third deformation event and subsequent fluid-rock interaction are the

manifestations of the dam impoundment or the result of the recurring seismicity of the last 50 years. However, the occurrences of prominent downdip slicken lines on shining slickensides with thin layers of unaltered pseudotachylites probably represent this deformation event as an ongoing process (Misra et al., 2022). Moreover, the partially dissolved plagioclase surface and biotitic remnants within neoformed chlorite reflect that the alteration processes are also underway and the transformation of the minerals is not completed. Besides, the

kinking observed in the neoformed chlorite in the microscopic study also demarcates that the stress build-up along the faults and fractures is still ongoing, as kinking is the expression of the shortening parallel to basal planes in response to the c-axis parallel strain.

In addition, clay minerals contribute significantly to the evolution of fluid pressure by absorbing or adsorbing water, which reduces the fault's shear strength and triggers the slip (Bruce, 1985; Vrolijk, 1990). The low frictional

strength of the clay minerals also results in a transient decrease in friction which accelerates the fault-weakening process and generates seismic slip (Van Der Pluijm, 2013). So, the accumulation of clay minerals along the faults and fractures, such as chlorite, illite etc., have a significant role in promoting slip in the Koyna-Warna Region (Piyal et al., 2021)

Actually, Chlorite consists of one octahedron sandwiched between two tetrahedra, which is alternated with

brucite-like interlayers. The oxygen atoms of the talc-like layer are connected to the hydroxyl ions of the brucite-like layer by weak hydrogen bonds in the chlorite structure. This anisotropic and weakly bonded layered crystal structure of chlorite prevents the dislocation glide from migrating to other planes (i.e., parallel to the c-axis or perpendicular to the layering) and restricts them only within the basal plane when shearing is applied parallel to the layers of chlorite (the basal or c planes) (Kronenberg et al., 1990). Therefore, in order to accommodate the

strain perpendicular to the c-axis, one plane of atoms moves over another without distorting in-plane bonds as a result of which atomic scale ripples are produced on the basal plane (Aslin et al., 2019). Thus, numerous small-scale ripplocations may occur with rising or continued stress, and they can store the strain energy until a threshold. When the stored energy reaches its yield, they may migrate with the energy gained and merge into kink bands curving the lattice to overcome the compelling force (Bell et al., 1986) **(Fig. 11)**. As a result, the yield strength of

chlorite rises proportionally with rising pressure up to the temperature at which it dehydrates, and behaves in a viscoelastic manner that is more favourable for the creep behaviour. In addition, relatively less frictional resistance ($\mu \sim 0.35$) of chlorite increases velocity-strengthening behaviour (An et al., 2021). Thus, chlorite crystal retains the strain energy within its lattice and prevents larger displacement of faults resulting in creep rather than earthquake slip. This kind of creep for long periods may result in numerous small earthquakes with less

devastation, as has also been seen in this region (more than 100,000 low-moderate magnitude earthquakes experienced during the past 5 decades).





**Figure 11.** Schematic diagram of the mechanism of secondary mineralisation and the response of chlorite under increasing stress conditions. (a) Simplified schematic diagram showing mass transfer of chemical components



600 during biotite chloritization followed by plagioclase dissolution and the formation of other associated secondary minerals (albite, calcite, illite and epidote) depending upon the remaining vacancies after chlorite formation (modified after Yuguchi et. al., 2015). (b) Schematic diagram showing how chlorite forms atomic scale ripples and develops kink bands in response to rising or continued stress- (I) Undeformed chlorite lattice, (II) small-scale ripplocations occurred with rising and continued stress and they migrate with the c-axis parallel strain energy,

605 (III) Numerous small-scale ripplocations merged and the elastic c-axis strain energy becomes permanent strain forming Kink bands.

  However, the mechanical response of epidote is radically different from that of chlorite. When temperature and pore fluid pressure are high enough during hydrothermal conditions, the frictional strength of epidote becomes significantly higher ($\mu \sim 0.73$) than that of chlorite (An et al., 2021). Moreover, with increasing

610 stress, the equant epidote grains tend to wear at the micron or submicron-scale asperity contacts, as a result of which fine particles are produced favouring velocity weakening behaviour. Hence, the presence of epidote in faults may promote subsurface instability, leading to unstable sliding.

  But, as has been discussed earlier, the formation of epidote is completely dependent upon the quantity of Al released during plagioclase dissolution as well as the remaining vacancies after the reduction of volume during

615 biotite chloritization. Hence, in our current study, the occurrence of epidote has not been identified during bulk mineralogical XRD whereas the rest of the minerals have been well documented, which led us to assume that the epidote production is relatively less than chlorite. Besides, chlorite is weaker than epidote and its relatively higher abundance in the faults and fractures disrupts the epidote-epidote contact asperities and prevents the wearing of epidote grains into fine particles. Thus, biotite chloritization in conjunction with epidote development along pre-

620 existing faults ultimately suppresses the velocity-weakening behaviour of epidote (Fagereng and Ikari, 2020). Mandal et al. (1998) have also encountered low seismicity and low-stress drop earthquakes between 1-4 km where we have found the presence of chlorite on the slip surfaces of the numerous small-scale faults as well as along the fractures. Similarly, Mahato and Shashidhar (2022) have found that the focal depths of the earthquakes from 1st May–25th June 2017 are mostly distributed at depths of 2.5–4.5 km in the Koyna-Warna Seismogenic Region but

625 surprisingly there is no evidence of seismicity within and around the depth segment of clay mineralisation, as found in this study (1-2 km) **(Fig. 12)**. Moreover, a low-stress drop value ($\Delta\sigma$) of 0.4 has also been evidenced in the new earthquake cluster (Mahato and Shashidhar, 2022). So, apart from the rupture directivity, source geometry, heterogeneity of micro-fractures, the presence of fluid along the faults and the strain-bearing crystal of chlorite produced due to fluid-rock interaction may also be responsible for such low median stress drop and

630 absence of significant seismicity at a particular depth.



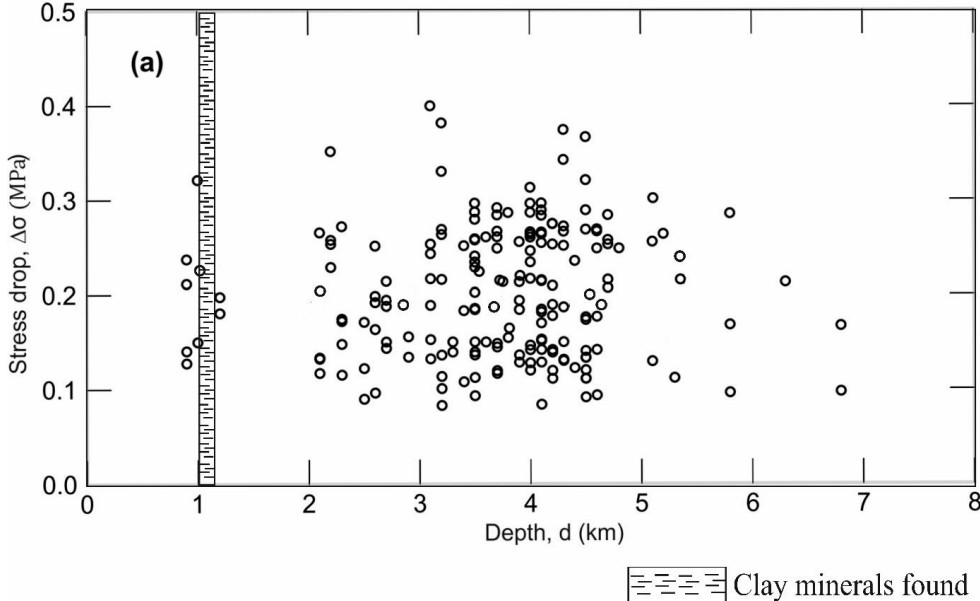

**Figure 12.** Stress drop (Δσ) vs. focal depth (d) for the new cluster of earthquakes. The diagram shows the lesser occurrence of seismicity at and around the depth where clay mineralisation has been reported in this study (Modified after Rao and Shashidhar (2016).


Therefore, the biotite chloritization along with the formation of a relatively smaller amount of epidote due to fluid-rock interaction along the subsurface faults may be beneficial for the Koyna-Warna Seismogenic Region, as their combined effect may promote steady fault creep and stabilise a fault by releasing the accumulated stress intermittently through the numerous small-scale tremors as observed in the past few decades. This gradual

stress release may be a boon since it shields the region from large earthquakes, which would otherwise be a bane to the area.

**7 Conclusion**

The study concludes-

(i) Fluid flow along the pre-existing fractures has resulted in the formation of

Chlorite+illite+epidote+albite+calcite in the basement rocks of the Deccan Trap Province in the Koyna-Warna Seismogenic Region representing propylitic grade of hydrothermal alteration.

(ii) Although the exact time of formation of the secondary minerals is quite difficult to assess but the fractures cutting plagioclase crystals as well as calcification and chloritization along those fractures surely establish that the fluid-rock interaction has taken place after the development of the fractures as a result of some brittle

deformation. The partially dissolved plagioclase surface and biotitic remnants within neoformed chlorite also reflect that the alteration processes are also underway and the transformation of the minerals is not completed.



Besides, the kinking observed in the neoformed chlorite also demarcates that the stress build-up along the faults and fractures is still ongoing.

(iii) Apart from the fault geometry and stress build-up due to reservoir impoundment, the clay mineralisation
along the faults and fractures, such as chlorite, illite etc., may have a connection with the seismicity in this region. The growth of chlorite may give rise to creep whereas epidote causes unstable sliding. However, chloritization may dominate epidote formation and result in continuous fault creep by releasing the accumulated stress through a series of small-scale tremors. This step-wise release of stress may be a boon for this region as it reduces the chances of large seismic activities, as also seen in recent times.





**Authors contributions**

**PH-** Conceptualisation, Methodology, Formal analysis, Investigation, Writing - Original Draft, Visualization, Data Curation. **MKS-** Matsyendra Kumar Shukla: Conceptualisation, Writing - Review & Editing, Supervision. **KK-** Writing - Review & Editing, Supervision. **AS-** Conceptualisation, Investigation, Resources, Writing - Review & Editing, Supervision, Project Administration.

**Competing interest**

The authors declare that they have no known competing financial interests or personal relationships that could have appeared to influence the work reported in this paper.

**Acknowledgements:**

        The authors are highly grateful to the Secretary of the Ministry of Earth Sciences as well as Dr. Arun Gupta,
Member-Secretary, SAGE Division, MoES, for providing the required funds to conduct this research [MoES/P.O.(Seismo)/1(374)/2019]. The authors express sincere gratitude to Dr. Sukanta Roy, Project Director, MoES-Borehole Geophysics Research Laboratory, Koyna, for his motivation and for supplying the precious core samples. Heartfelt thanks to Mr. Digant Vyas and all other scientific and technical staff of BGRL, for their cooperation during sample collection. Authors are overwhelmed with the cooperation of Dr. G.P. Gurumurthy,
Scientist-C, BSIP, for cooperation in ICP-OES analysis and Dr. Arvind Singh, Scientist-C & Dr. Adrita Chaudhuri, Scientist-B, BSIP, for providing facilities and suggestions for optical microscopic investigation. Thanks to Dr. Subodh Kumar (Technical Officer-D, FESEM laboratory), Mr. Jitendra Yadav (Technical Assistant-D, Sample Digestion Unit) and Dr. Amrit Pal Singh Chaddha (Technical Assistant-E, XRD). Sincere gratitude to the Director, BSIP, Lucknow, and Director, AcSIR, Ghaziabad, for the necessary facilities.



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
