# Peer review of "Fluid-rock interaction in the intraplate active seismic zone: Boon or bane?"

_EGUsphere, 2023_

## Author Comment (AC1)

Respected Sir/Ma'am,

We appreciate the time and effort that you have dedicated and are grateful for your insightful comments which have improved the quality of the manuscript as well as helped us to identify many silly mistakes. We are also sorry for such mistakes and will keep in mind in future. We are responding to all of your queries and concerns for your kind consideration. As there is no provision to upload the revised MS at this stage, the revised MS will be uploaded after the completion of the review process. We shall incorporate all of the suggestions and modifications in the revised MS. We are requesting you kindly consider our response. Your kind approval regarding the acceptance of the submitted MS will be highly appreciated. We shall be grateful for your kind perusal.

Regards,
Yours Sincerely,
Piyal Halder (First author) & Dr. Anupam Sharma (Corresponding author)

**RESPONSE TO THE SUGGESTIONS AND QUERIES**

**L14: XRD, acronyms should be introduced the first time you use the term.**

- Thanks for pointing out such a mistake. **The backronym of XRD i.e., X-Ray Diffraction** has been inserted in the revised manuscript which will be uploaded shortly after the completion of the review process.

**L16: I didn't find any discussion or interpretation to support the point about the acidic to neutral alteration conditions (pH 5.5-7) in the main text. How did you know the pH value?**

**L17: Similarly, the inferences about the alteration temperatures are missing.**

- Thanks for pointing out this flaw, otherwise it would mislead the readers. Actually, the temp. and pH mentioned in the text are characteristics of the propylitic grade of hydrothermal alteration (Fulignati, 2020). Hence, instead of mentioning a particular temp. and pH, a range has been provided. Relevant Citations have been already mentioned in the discussion part of the submitted MS file **(L498-L500)**. However, to avoid any confusion, the line relevant to this statement in the abstract section has been modified in the following manner-

"This indicates the fluid-rock interaction along these mechanically weak planes and subsequent propylitic grade of hydrothermal alteration **which characterizes** acidic to neutral conditions (pH 5.5-7) and the temperature of above 200-220 °C up to about 350 °C"

**L156-160: How was the transition from the faulting to normal faulting observed? Some references may need to be cited to support this transition and its potential relation with reservoir impoundment.**

- The evidence of fault transition from normal to strike-slip pattern and vice versa has been obtained from the inversion of 50 focal mechanism solutions of earthquakes that

occurred during the last 45 years in the Koyna–Warna region (Rao and Shashidhar, 2016). The relevant citation has been included in the text of the revised MS.

**L169-200: The authors introduced several faults in the study area. Which fault does the borehole pass through? It is also necessary to specify which part of the faults the samples came from, e.g., the principal slip zone or damage zone?**

- I am overwhelmed by your detailed review. These details are required for a reader to understand the entire research perspective. I am extremely sorry for such incompleteness in the information. The borehole KBH1 has cut numerous small-scale faults of the Koyna River fault Zone (KRFZ) which is already shown in **Fig. 1** of the submitted manuscript. Additionally, the same has been inserted in the sampling section of the revised MS, which will be uploaded later after the completion of the review process. The altered samples are collected from the slip surface of these small-scale faults (fault core) as mentioned in the **L192, L194, L206, and L286**. The same has also been inserted in the above-mentioned line for a clearer understanding as per your suggestions.

"The altered samples from the **slip planes of the small scale-faults** of the KRFZ have been collected through mesoscopic observations"

**Figure 3d: Is it fault gouge?**

- The figure is the mesoscopic observation of the sample KBH1_379 collected from 1073 m depth. The physical appearance as well as the presence of clay minerals, identified in the XRD **(Fig. 7c & d)**, infers that the sample may be a fault gouge or clayey gouge. Hence, the above-mentioned line has been modified in the revised MS as below-
"Clayey fault gouge reflecting the mechanical disintegration during fault slip followed by the clay formation due to fluid-rock interaction".

  The relevant line in the (L195-L196) main text has also been modified accordingly.

**L173, L177: Delete the full stop before the parenthesis of the citations. Please check throughout.**

- Thanks for pointing out such silly mistakes. Such mistakes have been noted and rectified as per your suggestions.

**L184: "litholog" → lithology. Please check throughout for errors.**

- Here, the "**litholog**" of the studied borehole has been shown at the side of the schematic diagram. The litholog exhibits the distribution of the porous and vesicular Deccan basalt upto 932 m depth which is underlain by the granitoid basement rocks.

**L270-271: Quartz with chess-board pattern generally reflects dislocation creep during high-temperature deformation. This mismatch with the presence of chlorite and epidote which indicates relatively low-temperature alteration.**

- I agree with your observations. However, the basement rocks in the study area have undergone three phases of deformation. The first of which corresponds to the

development of gneissosity in the granitic rocks, the second was the shearing events that caused pre-existing fabrics (S1) to be overprinted and transposed by the high-strain S2 fabrics, and the third was the emergence of anastomosing fractures as a function of brittle deformation such as faulting. The fractures cutting the plagioclase grains and fillings of chlorite and calcite throughout the network of fractures, as documented in our study (Fig. 4f, g), indicate that the percolation of fluid and subsequent formation or precipitation of secondary minerals along the fractures took place after the third deformation event in which faulting occurred and fractures were developed. So, the chessboard pattern of quartz as well as the marker offset within plagioclase indicates the third deformation event which gave rise to fractures. Later, fluid percolated through these fractures and resulted in the formation of secondary minerals. Hence, the presence of chlorite, illite as well as precipitation of calcite can't be correlated with the deformation pattern of quartz. (Please see **L556-L570** in the submitted MS)

**L463-478: Given the dissolution of plagioclase and potential release of Al, why not try TiO2 as the immobile element for the isocon plot?**

- The mass balance calculation and isocon plots have been executed following Grant (2005). The equation of an isocon is,

$$C_A = (M_O/M_A) \, C_O$$

$\quad$ [$C_A$= concentration of an element in altered rock

$\quad$ $C_O$= concentration of an element in unaltered rock

$\quad$ $M_O$= Mass before alteration

$\quad$ $M_A$= Mass after alteration]

Here, the $M_O/M_A$ is the slope of the isocon and the line with this slope indicates no gain and no loss in terms of mass. The equation is similar to the straight-line equation with zero intercept. So, this method employs the plotting of isocon in two different ways-

(i) The first method involves clustering $C_A$ and $C_O$ values along the y and x axis respectively. Later, the best-fit line is drawn through the origin by graphically searching the elements that lie on or near the line. This line is the isocon line and is named in the present study as the Best-fit isocon (Grant, 1986). So, there is no way to fit the isocon manually in this method. **However, if we see the best-fit isocons in all the altered samples, we notice that most of the best-fit isocons have passed near the TiO$_2$. So, it is evident that TiO$_2$ is almost immobile but not completely immobile.**

(ii) However, there is another method for drawing the isocon line, in which the line is extended by clustering and selecting the elements that show minimum gain/loss with respect to other elements, i.e. $[(C_A-C_o)/ C_o]$ will be minimum. This isocon has been named in the present study as the Immobility Isocon. For this purpose, an average of the concentrations of each element in all the unaltered samples ($C_o$) is calculated. Then gain or loss of that particular element in the altered sample is estimated with respect to the $C_o$. For example, the $C_o$ value of Al$_2$O$_3$ is 15.12 and for TiO2 is 0.39. In one altered sample, the concentration of Al$_2$O$_3$ is 21.34 and 1.46. So, the gain of Al$_2$O$_3$

and TiO$_2$ are 6.22 and 1.07 respectively. But with respect to C$_o$, there are gains of 0.41 and 2.73 for Al$_2$O$_3$ and TiO2 respectively **[Please see table S1 in the supplementary file]**. The same approach applies to other altered samples. In all the cases, Al$_2$O$_3$ has shown the minimum gain or loss with respect to C$_o$ **[Please see Table S1 in the supplementary file]. Hence, immobility isocon has been drawn mostly through Al$_2$O$_3$ rather than TiO$_2$. In the latter case (isocon drawing through TiO$_2$), a priori assumption is needed regarding the immobility of TiO$_2$. Such priori assumption may give misinformation regarding mass gain or loss. In fact, isocon drawn exactly through TiO$_2$ will show a loss of CaO in samples KBH1_378 and KBH1_379 which is contrary to the evidence of calcite precipitation. Similarly, in KBH1_381, if the isocon is drawn through TiO$_2$, it will represent the loss of CaO, MgO and FeO, which will contradict the findings of chlorite in XRD.**

**Moreover, this** method provides a holistic view of the mobility of elements throughout the geologic section or system (so an average is taken). Also, **Al activity in the hydrothermal fluid is restricted, it is mobile on the mineral scale, and it is immobile on a rock scale (Parneix et al., 1985). Most importantly, the findings emanated from immobility isocon match with the findings found in the case of the best-fit isocon method, which is completely statistically derived and doesn't involve any manual interference. That is why, the isocons have been drawn through Al$_2$O$_3$ is more reliable rather than through TiO$_2$.**

**L473: I am not going to agree that the precipitation of calcite is caused by the dissolution of plagioclase. As shown in Figure 6c, calcites are precipitated as calcite veins along fractures. This is generally considered to be related to precipitation from hydrothermal fluids. More evidence is required to indicate the source of calcite.**

I agree with your view that calcite is precipitated by the hydrothermal fluid. In our study, the heat derived from the faulting and fracturing must have contributed to this hydrothermal alteration (because the temperature at the depth of this study is less than 150 °C; See L501-L505 of the preprint file). So, there may be two sources of Ca in the hydrothermal fluid-

(i) One source is the overlying basalt which contains CaO of 8.76–11.7 wt% (Banerjee and Mondal, 2021). The meteoric water percolated through the basaltic rocks dissolves this Ca and transports it to the fracture or permeable faults, where the fluid deposits it in the form of calcite by degassing after heating.

(ii) Another source is the Ca$^{2+}$ released due to plagioclase dissolution in the granitoids. In this case, the Ca$^{2+}$ released due to plagioclase dissolution is mixed up with the descending meteoric water which gets hot by the heat generated by the faulting and fracturing and leads to the precipitation of calcite by degassing.

This is why, the probability of plagioclase dissolution as the source of calcite can't be ruled out. Hence, the above-mentioned line will be modified in a below-cited manner in the revised ms-

"Moreover, during this process along with Al$^{3+}$, Ca$^{2+}$ is also released which may act as one of the sources of calcite precipitated and elucidates the gain of CaO in the altered samples.

However, considering the CaO content (8.76–11.7 wt%) of the overlying basalt and the basalt's high weathering potential, the contribution of the Deccan Trap basalt in the introduction of $Ca^{2+}$ in the hydrothermal fluid must be taken into account"

**L647: "although" and "but" cannot be used at the same time. There are many grammar errors here and elsewhere, and authors have to check to ensure that the language used throughout the paper is clear, concise, and grammatically correct.**

- I am sorry for such carelessness. The same has been noted with care and will be rectified in the revised MS which will be submitted after the completion of the review.

**References:**

Banerjee, R. and Mondal, S.K., 2021. Petrology and geochemistry of the Deccan basalts from the KBH-7 borehole, Koyna Seismic Zone (Western Ghats, India): Implications for nature of crustal contamination and sulfide saturation of magma. Lithos, 380-381: 105864.

Fulignati, P., 2020. Clay Minerals in Hydrothermal Systems. Minerals, 10(10): 919.

Grant, J., 2005. Isocon analysis: A brief review of the method and applications. Physics and Chemistry of the Earth, Parts A/B/C, 30.

Parneix, J.C., Beaufort, D., Dudoignon, P. and Meunier, A., 1985. Biotite chloritization process in hydrothermally altered granites. Chemical Geology, 51(1): 89-101.

Rao, N. and Shashidhar, D., 2016. Periodic variation of stress field in the Koyna–Warna reservoir triggered seismic zone inferred from focal mechanism studies. Tectonophysics, 679.